

# A real-time approach to frequency-mixing spectroscopies: Application to sum and difference frequency generation in two-dimensional crystals

**Mike N. Pionteck$^{1\star}$, Myrta Grüning$^{2,3}$, Simone Sanna$^1$ and Claudio Attaccalite$^{3,4}$**

**1** Institut für Theoretische Physik and Center for Materials Research (LaMa),
Justus-Liebig-Universität Gießen, 35392 Gießen, Germany
**2** School of Mathematics and Physics, Queen's University Belfast,
Belfast BT7 1NN, United Kingdom
**3** European Theoretical Spectroscopy Facilities (ETSF)
**4** CNRS/Aix-Marseille Université, Centre Interdisciplinaire de Nanoscience de Marseille
UMR 7325 Campus de Luminy, 13288 Marseille cedex 9, France

$\star$ Mike.Pionteck@theo.physik.uni-giessen.de

## Abstract

We propose a computational framework to extract nonlinear response functions from real-time simulations in the presence of more than one external field. We apply this approach to the calculation of sum frequency generation (SFG) and difference frequency generation (DFG). SFG and DFG are second-order nonlinear processes where two lasers with frequencies $\omega_1$ and $\omega_2$ combine to produce a response at frequency $\omega = \omega_1 \pm \omega_2$. Compared with other nonlinear responses such as second-harmonic generation, SFG and DFG allow for tunability over a larger range. Moreover, the optical response can be enhanced by selecting the two laser frequencies in order to match specific electron-hole transitions. To assess the approach, we calculate the SFG and DFG of two-dimensional crystals, $h$-BN and MoS$_2$ monolayers, from real-time solution of an effective Schrödinger equation. Within the effective Schrödinger equation, one can select from various levels of theory for the effective one-particle Hamiltonian to account for local-field effects and electron-hole interactions. We compare results obtained within the independent particle picture and including many-body effects. Such comparison allows us to identify and characterize excitonic features in the obtained spectra. Additionally, we demonstrate that our approach can also extract higher-order response functions, such as field-induced second-harmonic generation. We provide an example using the $h$-BN bilayer.

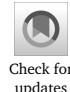

## Contents



# 1 Introduction

Sum frequency generation (SFG) and difference frequency generation (DFG) spectroscopy are powerful experimental techniques where the spectrum is the second-order nonlinear optical response $\chi^{(2)}$ resulting from the combination of two laser fields (see Fig. 1(a) and (b)). These techniques are highly sensitive to surfaces and interfaces [1–3]. In recent years, there has been a growing interest in the application of SFG/DFG in condensed matter physics. In particular, SFG/DFG was reported in layered MoS$_2$ and related heterosystems using either band-filtered supercontinuum illumination [4,5] or wavelength-dependent spectroscopy [5–8]. More interestingly, one can make the SFG dual resonant with the exciton, strongly enhancing its response function, as shown recently in two-dimensional (2D) materials [9]. Then, SFG can serve to explore exciton-exciton transitions as an alternative and complementary technique to pump and probe spectroscopy [10]. The design and interpretation of such experiments call for the development of theoretical approaches that can, for a specific material, capture both the nonlinear light-matter interaction and the many-body physics of excitons.

So far, few theoretical studies have been reported in the literature on SFG and DFG in solids. The SFG was investigated using either simple two-band models [11] or from first-principles, using the Greenwood-Kubo formalism, within an independent particle picture [9]—thus missing excitonic effects. In this work, we put forward a general approach to extract the SFG and DFG spectra from real-time simulations. We implement this approach within the first-principles framework of Ref. [12] in which the coupling of the electrons with the external electric field based on the Berry-phase formulation of the dynamical polarization [13] and many-body effects, including excitonic effects, are accounted for through an effective Hamiltonian [14]. We apply the approach to calculate the SFG and DFG of MoS$_2$ and $h$-BN monolayers, both within the independent particle picture and including excitonic effects. The approach presented is not limited to the SFG and DFG but allows an efficient calculation of other response functions, such as field-induced second-harmonic generation (FI-SHG) [15,16]. FI-SHG involves applying an electric field, such as laser pulses, direct current (DC), or an intense terahertz (THz) electric field, to a crystal and measuring the resulting second-harmonic intensity, which can provide important insights into the material properties. Centrosymmetric crystals, which

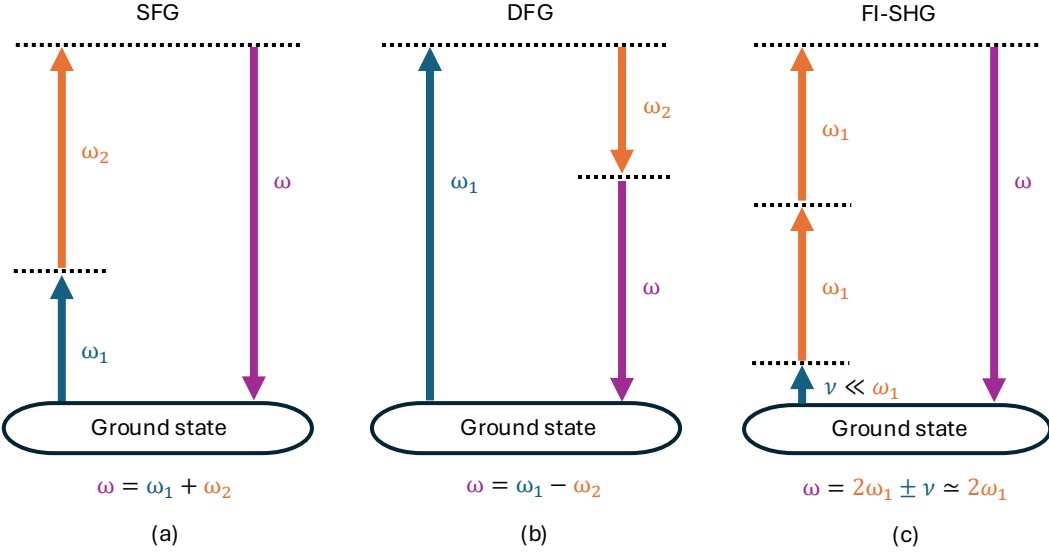

Figure 1: A schematic representation of the nonlinear processes studied in this work: (a) sum frequency generation (SFG), (b) difference frequency generation (DFG) and (c) field-induced second-harmonic generation (FI-SHG).

have a null second-harmonic response (in the dipole approximation), are of particular interest because the applied field breaks the symmetry and produces even-order harmonic radiation. For a static electric field a real-time approach to study FI-SHG has been proposed in Ref. [17]. Here we go a step further, and put forward a framework to simulate FI-SHG in the presence of time-dependent pump fields. For a $h$-BN bilayer, which is a centrosymmetric crystal, we calculate the second-harmonic response induced by a THz field breaking the inversion symmetry [Fig. 1(c)].

The manuscript is organized as follows. In Sec. 2, we present the real-time approach used to obtain the dynamical polarization (Sec. 2.1) and provide a description of SFG, DFG, and FI-SHG through their Lehmann representations (Secs. 2.2-2.3). In Sec. 3, we detail and contrast the signal process techniques (discrete Fourier analysis and least squares optimization) used to obtain the SFG/DFG and FI-SHG. After outlining the computational details in Sec. 4, we discuss in Sec. 5 the results obtained for the SFG and DFG of MoS$_2$ and $h$-BN monolayers and for the FI-SHG in the $h$-BN bilayer.

## 2 Theoretical background

We obtain the nonlinear optical susceptibilities from the time evolution of Bloch electrons in a uniform time-dependent electric field [12]. We extend the approach developed in Ref. [12] that was limited to a single monochromatic field in Sec. 2.1. This extension allows us to access a range of nonlinear phenomena. We focus on the SFG and DFG (Sec. 2.2) and the FI-SHG (Sec. 2.3).

### 2.1 Nonlinear response from real-time simulations

The time evolution of the electronic system induced by two monochromatic homogeneous fields $\mathcal{E}_1(t)$ and $\mathcal{E}_2(t)$ is described by the following equation of motions (EOMs) for the valence

Bloch states:

$$i\hbar \frac{d}{dt}|v_{m\mathbf{k}}\rangle = \left\{ H_{\mathbf{k}}^{\mathrm{MB}} + ie\left[\boldsymbol{\mathcal{E}}_1(t) + \boldsymbol{\mathcal{E}}_2(t)\right]\cdot \tilde{\partial}_{\mathbf{k}} \right\}|v_{m\mathbf{k}}\rangle, \tag{1}$$

where $|v_{m\mathbf{k}}\rangle = |v_{m\mathbf{k}}(t)\rangle$ is the periodic part of the time-dependent Bloch states. On the right-hand side of Eq. (1), the second term describes the coupling with the external field in the dipole approximation. The coupling takes the form of a $\mathbf{k}$-derivative operator. The tilde indicates that the operator is gauge covariant, i.e. the solutions of Eq. (1) do not change under unitary rotation at $\mathbf{k}$ (see Ref. [13] for more details). $H_{\mathbf{k}}^{\mathrm{MB}}$ is the effective many-body Hamiltonian. Different correlation effects can be accounted for by constructing the corresponding effective Hamiltonian. The simplest approximation for the effective Hamiltonian we consider in this work is the independent particle approximation (IPA) describing the unperturbed (zero-field) Kohn-Sham system [18]

$$H^{\mathrm{MB}} \equiv H^{\mathrm{KS}} = -\frac{\hbar^2}{2m}\sum_i \nabla_i^2 + V_{\mathrm{eI}} + V_{\mathrm{h}}[\rho^0] + V_{\mathrm{xc}}[\rho^0] = \sum_i \varepsilon_{n,\mathbf{k}}\left|u_{n,\mathbf{k}}\right\rangle\left\langle u_{n,\mathbf{k}}\right|, \tag{2}$$

where $\hat{V}_{\mathrm{eI}}$ is the electron-ion interaction, $\hat{V}_{\mathrm{H}}[\rho^0]$ and $\hat{V}_{\mathrm{xc}}[\rho^0]$ are respectively the Hartree and exchange-correlation potentials evaluated at the unperturbed ground-state electron density $\rho^0$, $\varepsilon_{n,\mathbf{k}}$ the Kohn-Sham energies (eigenvalues) and $|u_{n,\mathbf{k}}\rangle$ the periodic part of the Kohn-Sham wavefunctions. The Hamiltonian in Eq. (2) has two main shortcomings. First, the band gaps obtained from the Kohn-Sham energies are systematically underestimated—yielding to an overall underestimation of the optical gap. Second, the response of the density functional potentials to the change in the density is missing and so are local-field effects and quasi-particle excitations. The former shortcoming is corrected by introducing a state-dependent scissor operator term,

$$\Delta \hat{H}_{\mathbf{k}} = \sum_n \Delta_{n\mathbf{k}}\left|u_{n,\mathbf{k}}\right\rangle\left\langle u_{n,\mathbf{k}}\right|, \tag{3}$$

where state-dependent $\Delta_{n\mathbf{k}}$ is the scissor operator. That is, considering the Kohn-Sham energies as the zero-order approximation to quasi-particle energies, the scissor operator introduces a first-order correction. Such corrections can be obtained from first-principles within the so-called *GW* approximation derived from many-body perturbation theory [19] (a simpler approach is to have a single scissor operator chosen to reproduce e.g. the experimental fundamental band gap). To address the second shortcoming, we include the fluctuation of the Hartree potential (Eq. (2)) generated by the variation of the electron density $\Delta\rho = \rho - \rho^0$ induced by the external fields [12], and the screened-exchange self-energy term $\Sigma_{\mathrm{SEX}}[\Delta\gamma]$ where $\Delta\gamma$ is the fluctuation of the density matrix [20]. This latter term is a nonlocal operator which accounts for the screened electron-hole attraction and introduces excitonic effects. We so obtain the following effective Hamiltonian,

$$H_{\mathbf{k}}^{\mathrm{MB}} \equiv H_{\mathbf{k}}^{\mathrm{KS}} + \Delta H_{\mathbf{k}} + V_{\mathrm{h}}(\mathbf{r})[\Delta\rho] + \Sigma_{\mathrm{SEX}}[\Delta\gamma], \tag{4}$$

which is the most accurate Hamiltonian considered in this work and is known as time-dependent adiabatic GW (TD-aGW)[1] approximation [20]. This Hamiltonian (Eq. (4)) is built in such a way to reproduce the band structure and the response functions of the standard *GW* + BSE approach [19, 22]. This has been shown in the limit of small perturbation both analytically and numerically in Ref. [20].

From the solutions $|v_{m\mathbf{k}}\rangle$ in Eq. (1), we calculate the real-time polarization along the lattice vector $\mathbf{a}$ as [13]:

$$\mathcal{P}_{\parallel}(t) = -\frac{ef|\mathbf{a}|}{2\pi V}\mathrm{Im}\log \prod_{\mathbf{k}}^{N_{\mathbf{k}}-1} \det S(\mathbf{k}, \mathbf{k}+\mathbf{q}; t), \tag{5}$$

---

[1]Notice that the TD-aGW approximation was called TD-BSE, TD-SEX or TD-HSEX in previous publications [10, 20, 21].

where $S(\mathbf{k}, \mathbf{k} + \mathbf{q}; t)$ is the overlap matrix between the time-dependent valence states $|v_{n\mathbf{k}}\rangle$ and $|v_{m\mathbf{k+q}}\rangle$, $V$ is the unit cell volume, $f$ is the spin degeneracy, $N_{\mathbf{k}}$ is the number of $\mathbf{k}$-points along the polarization direction, and $\mathbf{q} = 2\pi/(N_{\mathbf{k}}\mathbf{a})$. Then, the $n$-order susceptibilities $\chi^{(n)}$ are extracted from the frequency-dependent polarization expanded in a power series of the incident fields as:

$$
\mathcal{P}_\alpha(\omega) = \sum_{\beta=1}^{3}\sum_{i=1}^{2}\chi_{\alpha\beta}^{(1)}(\omega;\omega_i)\mathcal{E}_\beta(\omega_i) + \sum_{\beta,\gamma=1}^{3}\sum_{i,j=1}^{2}\chi_{\alpha\beta\gamma}^{(2)}(\omega;\omega_i,\omega_j)\mathcal{E}_\beta(\omega_i)\mathcal{E}_\gamma(\omega_j)
$$
$$
+ \sum_{\beta,\gamma,\delta=1}^{3}\sum_{i,j,k=1}^{2}\chi_{\alpha\beta\gamma\delta}^{(3)}(\omega;\omega_i,\omega_j,\omega_j)\mathcal{E}_\beta(\omega_i)\mathcal{E}_\gamma(\omega_j)\mathcal{E}_\delta(\omega_k) + O(\mathcal{E}^4),
\tag{6}
$$

where $\omega_i, \omega_j$ are frequencies of the perturbing fields $\mathcal{E}_\beta, \mathcal{E}_\gamma$ and $\omega$ the frequency of the outgoing polarization, with $\alpha, \beta, \gamma$ denoting the Cartesian directions.

## 2.2 Sum/difference frequency generation

In Fig. 1(a),(b) we present a schematic representation of the SFG and DFG response functions studied in this work which correspond to $\chi^{(2)}(\omega, \omega_i, \omega_j)$ in Eq. (6) with $\omega = \omega_i \pm \omega_j$. In Sec. 3.2, we explain how the SFG and DFG are derived from the time-dependent polarization. Here, we discuss their Lehmann representation which is useful in the interpretation of the results.

The general form of the second-order susceptibility, $\chi_{\alpha\beta\gamma}^{(2)}(\omega_3; \omega_1, \omega_2)$ in the Lehmann representation, obtained through second-order perturbation theory [23] reads:

$$
\chi_{\alpha\beta\gamma}^{(2)}(\omega_3;\omega_1,\omega_2) = \frac{-ie^3}{m^3\tilde{\omega}_3\tilde{\omega}_1\tilde{\omega}_2}\sum_{\lambda\lambda'}\Bigg[\frac{\alpha_{0\lambda}\beta_{\lambda\lambda'}\gamma_{\lambda'0}}{(\tilde{\omega}_2-\Omega_{\lambda'})(\tilde{\omega}_3-\Omega_\lambda)} + \frac{\beta_{0\lambda}\gamma_{\lambda\lambda'}\alpha_{\lambda'0}}{(\tilde{\omega}_1+\Omega_\lambda)(\tilde{\omega}_3+\Omega_{\lambda'})}
$$
$$
- \frac{\gamma_{0\lambda}\alpha_{\lambda\lambda'}\beta_{\lambda'0}}{(\tilde{\omega}_1-\Omega_{\lambda'})(\tilde{\omega}_2+\Omega_\lambda)} + \frac{\alpha_{0\lambda}\gamma_{\lambda\lambda'}\beta_{\lambda'0}}{(\tilde{\omega}_1-\Omega_{\lambda'})(\tilde{\omega}_3-\Omega_\lambda)}
\tag{7}
$$
$$
+ \frac{\gamma_{0\lambda}\beta_{\lambda\lambda'}\alpha_{\lambda'0}}{(\tilde{\omega}_2+\Omega_\lambda)(\tilde{\omega}_3+\Omega_{\lambda'})} - \frac{\beta_{0\lambda}\alpha_{\lambda\lambda'}\gamma_{\lambda'0}}{(\tilde{\omega}_2-\Omega_{\lambda'})(\tilde{\omega}_1+\Omega_\lambda)}\Bigg],
$$

where $\Omega_\lambda$ are the excitation energies of the system, and $\alpha_{\lambda\lambda'}$ refers to momentum matrix elements $\langle\lambda|P_\alpha|\lambda'\rangle$ between two excited states and similarly for $\beta_{\lambda\lambda'}$ and $\gamma_{\lambda\lambda'}$. Here, $\tilde{\omega}_1 = \omega_1 + i\eta$, $\tilde{\omega}_2 = \omega_2 + i\eta$ and $\tilde{\omega}_3 = \tilde{\omega}_1 + \tilde{\omega}_2$. $\mathbf{P}$ is the many-body momentum operator, i.e. $\mathbf{P} = \sum_i \mathbf{p}_i$ where $\mathbf{p}_i$ is the single-particle momentum operator acting on particle $i$ and $\eta$ is a small positive number that introduces dephasing/dissipation effects. An approximation of Eq. (7) can be derived by replacing many-body states and energies with excitonic ones [22]: $|\lambda\rangle \simeq |\Psi_\lambda^{\text{exc}}\rangle$ and $\Omega_\lambda \simeq E_\lambda$. When we consider the SFG case $\omega_3 = \omega_1 + \omega_2$, we retain only positive contributions to the $\chi^{(2)}$ and we get

$$
\chi_{\alpha\beta\gamma}^{(2)}(\omega_1+\omega_2;\omega_1,\omega_2) \approx \sum_{\lambda\lambda'}\Bigg[\frac{P_{\alpha,0\lambda}P_{\beta,\lambda\lambda'}P_{\gamma,\lambda'0}}{(\tilde{\omega}_2-E_{\lambda'})(\tilde{\omega}_1+\tilde{\omega}_2-E_\lambda)} - \frac{P_{\gamma,0\lambda}P_{\alpha,\lambda\lambda'}P_{\beta,\lambda'0}}{(\tilde{\omega}_1-E_{\lambda'})(\tilde{\omega}_2+E_\lambda)}
$$
$$
+ \frac{P_{\alpha,0\lambda}P_{\gamma,\lambda\lambda'}P_{\beta,\lambda'0}}{(\tilde{\omega}_1-E_{\lambda'})(\tilde{\omega}_1+\tilde{\omega}_2-E_\lambda)} - \frac{P_{\beta,0\lambda}P_{\alpha,\lambda\lambda'}P_{\gamma,\lambda'0}}{(\tilde{\omega}_2-E_{\lambda'})(\tilde{\omega}_1+E_\lambda)}\Bigg],
\tag{8}
$$

where $\lambda$ now indicates the excitonic state and $P_{\alpha,\lambda\lambda'} = \langle\Psi_\lambda^{\text{exc}}|p_\alpha|\Psi_{\lambda'}^{\text{exc}}\rangle$ [10]. A similar procedure has been applied in the literature for the second-harmonic generation (SHG) case [24, 25]. From this formula we could expect strong peaks when $\omega_1 + \omega_2$ is resonant with the excitonic energy or when single laser frequencies $\omega_1, \omega_2$ are resonant with an exciton. Note that the

first and third terms have poles at both one-photon (e.g. $\omega_1, \omega_2$) and two-photon ($\omega_1 + \omega_2$) resonances. Finally, if $P_{\alpha,\lambda\lambda'}$ is different from zero for $\lambda \neq \lambda'$, there may also be resonances with two distinct excitonic energies.

## 2.3 Field-induced second-harmonic generation

Among higher-order responses that can be extracted from Eq. (6), we look at the FI-SHG.

Let's consider a system with no second-harmonic response. In presence of an external field, the polarization in a given direction acquires contributions of the form,

$$\mathcal{P}(2\omega^{\pm}) = \hat{\chi}^{(3)}(2\omega \pm \nu; \nu, \omega, \omega)\mathcal{E}_1(\nu)\mathcal{E}_2^2(\omega). \tag{9}$$

If $\mathcal{E}_1$ is static, i.e. $\nu = 0$, the third-order susceptibility extracted from this contribution to the polarization gives the FI-SHG. Similarly, if $\nu \ll \omega$, i.e $\mathcal{E}_1$ is in the THz range, we have $2\omega \pm \nu \approx 2\omega$ and the expression in Eq. (9) can be rewritten as,

$$\mathcal{P}(2\omega^{\pm}) \approx \hat{\chi}^{(3)}(2\omega; \nu, \omega, \omega)\mathcal{E}_1(\nu)\mathcal{E}_2^2(\omega). \tag{10}$$

Then, extracting the $\chi^{(3)}$ from $\mathcal{P}(2\omega^+)$ or $\mathcal{P}(2\omega^-)$, one obtains the corresponding FI-SHG for low-frequency time-dependent pump fields.

# 3 Signal processing: Nonlinear susceptibilities

Adding an extra field to the formalism presented in Ref. [12] to obtain Eq. (1) is straightforward. The challenging part, and the main contribution of this work, is finding feasible, accurate strategies for extracting the relevant nonlinear susceptibilities from the resulting polarization $\mathcal{P}(t)$. A strategy based on the discrete Fourier transform (FT) is used in Ref. [12] (Sec. 3.1) for the case of one external monochromatic field. This analysis can be extended to more external monochromatic fields (Sec. 3.2), but as the common period for two, or more, (commensurate) frequencies can be of several hundreds of fs, this implies very long and computationally costly simulations. We thus propose an alternative strategy based on the least squares fit (LSF), which turns out to be as accurate as the discrete FT without requiring too long simulations. The method proposed in this section is quite general and could be applied to Hamiltonians that differ from the one used in this work. Examples include tight-binding and other lattice models.

## 3.1 One external monochromatic field

As described in Ref. [12], the harmonic generation is obtained by the simulation of a system under the effect of an external periodic electric field with frequency $\omega_L$, $\mathcal{E}(t) = \mathcal{E}_0 \sin(\omega_L t)\Theta(t)$. A dephasing term is added to the Hamiltonian [12] so that for simulation times larger than the dephasing time $1/\gamma_{\text{deph}}$, the resulting time-dependent polarization can be written as:[2]

$$\mathcal{P}(t) = \sum_{n=-\infty}^{\infty} \mathbf{C}^{(n)} \exp\left[-i\omega^{(n)}t\right], \tag{11}$$

with $\omega^{(n)} = n\omega_L$. The complex Fourier components $\mathbf{C}^{(n)}$ can be determined by truncating the Fourier series to an order $S$ and sampling $2S + 1$ values $\mathcal{P}_i \equiv \mathcal{P}(t_i)$ within a period

---

[2]In general, the signal is sampled after a time which is about 6 times the dephasing time of the system. Then, the (spurious) term in the response functions arising from the sudden switch-on of the external field is negligible.

$T_L = 2\pi/\omega_L$. For each direction $\alpha$, this yields

$$\sum_{n=-S}^{S} \mathcal{F}_i^{(n)} C_\alpha^{(n)} = \mathcal{P}_{\alpha,i}, \qquad i = 1, 2S + 1, \tag{12}$$

where $\mathcal{F}_i^{(n)} \equiv \exp\left[-i\omega^{(n)} t_i\right]$. The solution of the $(2S+1)$ system of linear equations [Eq.(12)] outputs the $C_\alpha^{(n)}$, from which in turn one gets the $n^{\text{th}}$-order susceptibility by dividing by the $n^{\text{th}}$ power of the $\alpha$ component of $\mathcal{E}_0$. The nonlinear susceptibilities converge rapidly with $S$: in Ref. [12] it was found that second-order susceptibilities converge already with $S = 4$ and third-order susceptibilities with $S = 6$.

## 3.2 Two external monochromatic fields

We consider two monochromatic fields, with commensurate frequencies, $\omega_1$ and $\omega_2$. Frequencies are commensurate as long as they are rational numbers (that is $\omega_1, \omega_2 \in \mathbf{Q}$), which is the case in numerical implementations. The greatest common divisor of $\omega_1$ and $\omega_2$ leads to the fundamental frequency,

$$\omega_0 = \frac{\gcd(\lfloor 10^m \omega_1 \rfloor, \lfloor 10^m \omega_2 \rfloor)}{10^m}, \tag{13}$$

where $m = \max(n_1, n_2)$ and $n_1$ and $n_2$ are the number of decimals in $\omega_1$ and $\omega_2$, respectively.

The resulting polarization $\mathcal{P}(t)$ is periodic with period $T = 2\pi/\omega_0$ and hence can be written as a Fourier series as in Eq. (11), in terms of the harmonics of $\omega_0$. As the frequencies of the external fields are multiple of the fundamental frequency $\omega_1 = M\omega_0$ and $\omega_2 = N\omega_0$, the $|M \pm N|$ harmonics correspond to the SFG/DFG. When it comes to setting up the system of linear equations (Eq. (12)), the sum over the harmonics must be truncated to an appropriate $S$ to include these processes (that is $S \gtrsim M + N$). When compared with a single external field, the dimension of the system of linear equations is significantly larger. More critically, $T$ can be orders of magnitude larger than typical laser periods in the near-infrared to near-UV range

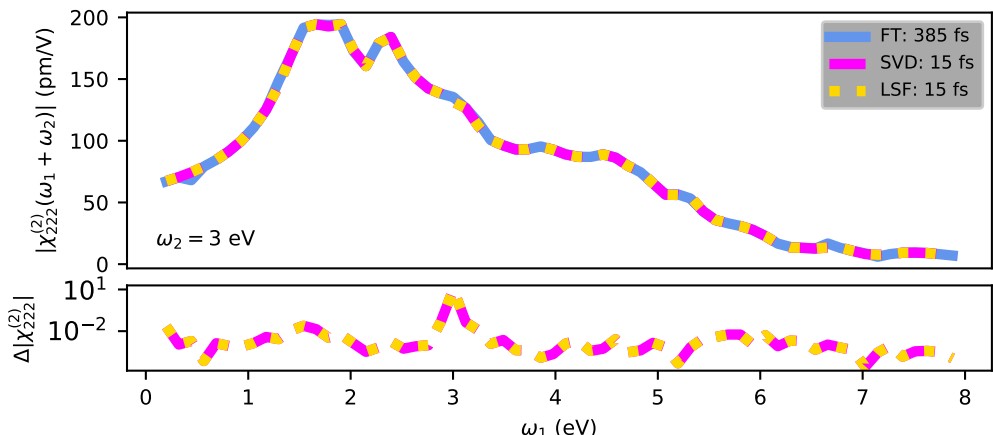

Figure 2: SFG of $h$-BN with a pump frequency $\omega_2 = 3$ eV obtained at the IPA level using the full discrete Fourier transformation (FT) (blue solid line), singular value decomposition (SVD) (magenta dashed line), and the least square fit (LSF) (yellow dotted line). Results obtained with different sampling times are shown. The discrete FT needs a sampling time (385 fs) about 26 times larger than the SVD and LSF (15 fs). The bottom panel shows the difference in logarithmic scales for SVD and LSF, respectively, with 5 fs less sampling time to show the convergence.

($\approx 1-5$ fs) for any pair of frequencies with more than one decimal leading to computationally intensive simulations (see Fig. 2).

Alternatively, we can expand the polarization $\mathcal{P}(t)$ as the product of two Fourier series, one in terms of the harmonics of $\omega_1$ and the other in terms of the harmonics of $\omega_2$,

$$\mathcal{P}(t) = \sum_{n,m=-\infty}^{\infty} \mathbf{C}^{(n,m)} \exp\left[-i(n\omega_1 + m\omega_2)t\right], \tag{14}$$

where the matrix of the Fourier coefficients is denoted as $\mathbf{C}^{(n,m)}$. The Fourier coefficients can be found by the solution of the system of $(2S_1 + 1)(2S_2 + 1)$ linear equations

$$\sum_{n=-S_1}^{S_1} \sum_{m=-S_2}^{S_2} \mathcal{F}_i^{(n,m)} C_\alpha^{(n,m)} = \mathcal{P}_{\alpha,i}, \tag{15}$$

where $\mathcal{F}_i^{(n,m)} \equiv \exp[-i(n\omega_1 + m\omega_2)t_i]$ and $S_1, S_2$ are the maximum number of harmonics considered for each external field. Compared to the discrete FT with one external field, it is straightforward to identify the relevant coefficients, for example, $n = 1, m = 1$ gives the SFG and $n = 2, m = \pm 1$ the FI-SHG (choosing $\omega_2 = \Omega$). On the other hand, generally, the system of linear equations in Eq. (15) is ill-conditioned. To solve Eq. (15), we calculate the Moore-Penrose inverse [26, 27], also called pseudoinverse, of $\mathcal{F}_i^{(n,m)}$ by using the singular value decomposition (SVD). This approach (that we will refer to as SVD) allows to work with a much smaller sampling time than $T$, so to significantly reduce the time from the simulations when compared with the discrete FT.

A further approach to obtain the Fourier coefficients is by solving a least squares problem [28]. This consists in finding the set of $C_\alpha^{(n,m)}$ that minimizes:

$$\sum_{i=1}^{N} |\mathcal{P}_\alpha(t_i) - \bar{\mathcal{P}}_\alpha(t_i)|^2, \tag{16}$$

where $N$ is the number of sampling points and $\bar{\mathcal{P}}$ is the Fourier series of Eq. (14) truncated to an order of $S$. Similarly to the SVD-based approach, the main advantage of the LSF over the discrete FT is that it is sufficient to sample part of the period $T$ to find the Fourier coefficient accurately. For instance, for $h$-BN, Fig. 3 compares the polarization from the simulation with that reconstructed from Eq. (14) with the coefficients obtained from LSF. Although only 15 fs are sampled, the LSF provides an accurate result.

The accuracy and performance of the three approaches are contrasted in Fig. 2 for the SFG of $h$-BN. As can be seen, SVD and LSF provide the same solution for the same sampling time due to the linear dependence between the components of polarization and Fourier matrix. In a more general nonlinear problem this is not necessarily the case. According to our tests, SVD and LSF are hence equally efficient. Using the approaches based on the SVD and the LSF allows to cut simulation time by a factor 26 compared to the discrete FT.

### 3.2.1 Further numerical considerations

For the LSF to be accurate, one must carefully sample the key features of the signal, e.g. minima, maxima, and turning points. This implies that, in general, the LSF needs more sampling points $\mathcal{P}_{\alpha,i}$ (that is a higher sampling rate) than the discrete FT. This does not impact the cost of the simulations since a small time step is needed to integrate the equation of motion in Eq. (1). We found that non-uniform sampling (logarithmic or randomized) is more effective—thus reducing the sampling rate—than uniform sampling in capturing the key features of the

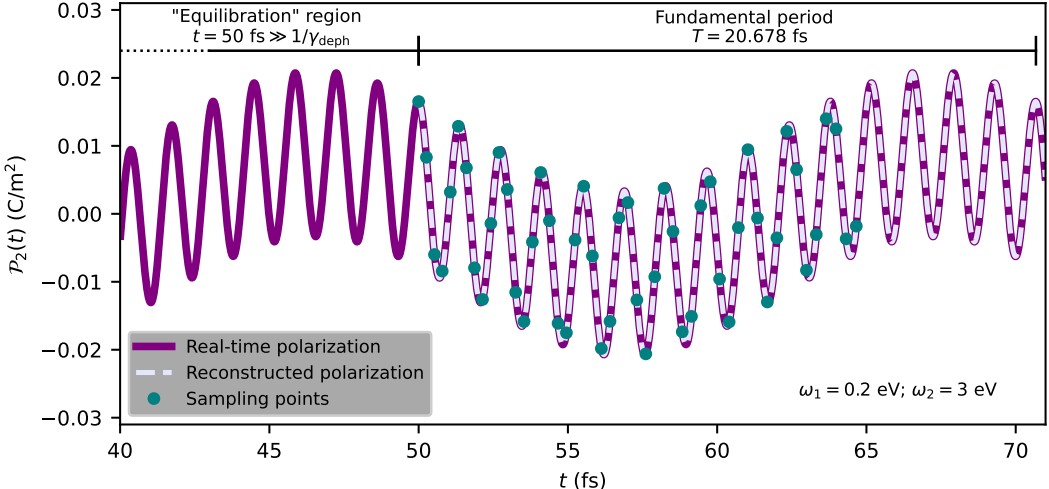

Figure 3: The time-dependent polarization (purple solid line) of $h$-BN calculated at the independent particle level with two electric fields ($\omega_1 = 0.2$ eV, $\omega_2 = 3$ eV). The signal can be divided into two regions: an initial "equilibration" region (up to $t \gg \gamma_{\text{deph}}$, here $t = 50$ fs) during which the system's eigenfrequencies are suppressed by dephasing and a region where Eq. (14) holds. In the second region the polarization is logarithmically sampled (teal dots) within a converged time window of 15 fs, smaller than the fundamental period of 20.678 fs of the signal. This sampling time is sufficient to correctly determine the Fourier coefficients by the least square fit (LSF) as verified by reconstructing the polarization (lavender dashed line) within the fundamental period using the Fourier coefficients and the truncated Eq. (14).

signal, especially for high-frequency signals. An example of logarithmic sampling for LSF is shown in Fig. 3.

Another aspect to consider is which element of the susceptibility tensor to calculate. Depending on the crystal symmetry, certain tensor elements are equivalent; however, the directions in which the linear response is non-zero, tend to be less precise and more unstable. This is due to two factors. First, the spurious signal arising from the sudden switch-on is stronger. Second, the fitting procedure is less accurate since the linear response coefficient is about 6 orders of magnitude larger than the SHG, SFG and DFG ones. Therefore, if possible, one must choose the directions where the linear response is absent. For example, in the case of the second-order response for 2D hexagonal systems such as those studied here, it is more convenient to study the off-diagonal elements, e.g., $\chi^{(2)}_{122}$, given that $\chi^{(2)}_{122} = \chi^{(2)}_{222}$ by crystal symmetry.

Furthermore, a worst case is when $\omega_1 \cong \omega_2$. This increases the necessary simulation time a lot for the discrete FT. However, also for SVD and LSF this case is challenging as indicated in Fig. 2 by the peak of the logarithmic difference between different sampling times at around 3 eV ($\omega_1 \cong \omega_2$). When they are exactly equal, the least square problem is ill-posed. For example, if we write down the first term of the second-order polarization:

$$\mathcal{P}^{(2)} = \chi^{(2)}(2\omega_1; \omega_1, \omega_1)\mathcal{E}^2(\omega_1) + \chi^{(2)}(2\omega_2; \omega_2, \omega_2)\mathcal{E}^2(\omega_2)$$
$$+ \chi^{(2)}(\omega_1 + \omega_2; \omega_1, \omega_2)\mathcal{E}(\omega_1)\mathcal{E}(\omega_2) + \ldots$$

and set $\omega_1 \cong \omega_2$, SHG and SFG become equivalent, i.e. their coefficients cannot be fixed using least square optimization. Similar issues arise if one of the frequencies is an integer multiple of the other. In those cases, a few strategies may be adopted. One can eliminate the

Table 1: All the parameters used in the nonlinear response calculations for both $MoS_2$ and $h$-BN monolayers: the **k**-point sampling used in the IPA (TD-aGW in parentheses), the range of bands considered, the cut-off, $\epsilon_{cut}$, and the number of bands, $\epsilon_{bands}$, used to converge the dielectric function $\epsilon_{\mathbf{G},\mathbf{G}'}$, the value of shift ($\Delta E_{so}$) for the scissor operator applied to the Kohn-Sham band structure, the height of the supercell, $L_z$ and the effective layer thickness, $d_{eff}$. For the 2L-$h$-BN calculations are only at IPA level, so no information about dielectric constant and scissor operator are reported.

| System | **k**-points | $N_b$ | $\epsilon_{cut}$(Ha) | $\epsilon_{bands}$ | $\Delta E_{so}$ (eV) | $L_z$ (Å) | $d_{eff}$ (Å) |
|--------|--------------|-------|----------------------|--------------------|----------------------|-----------|---------------|
| $MoS_2$ | $30 \times 30$ ($21 \times 21$) | 4-13 | 5 | 200 | 0.72 | 10.88 | 6.15 |
| $h$-BN | $30 \times 30$ ($18 \times 18$) | 3-7 | 5 | 200 | 3.35 | 10.58 | 3.33 |
| 2L-$h$-BN | $30 \times 30$ | 5-14 | - | - | - | 10.58 | 6.66 |

repeated terms in the fitting function, or use as starting values for the optimization those of the closest frequency already calculated, or simply interpolate the result from the neighbouring frequencies.

Note that the quasi-degenerate case is also the worst case for the discrete FT since closely spaced frequencies result in rapid beats in the signal. To accurately capture these beats, a very long sampling range is necessary to resolve the beat frequency ($\omega_1 - \omega_2$). Ultimately, cases involving degenerate or quasi-degenerate frequencies are of little theoretical and experimental interest. Theoretically, for degenerate frequencies, the procedures in Ref. [12] (see Sec. 3.1) can extract the second-harmonic, while experimentally the interest lies in distinct frequencies resonant with different electron-hole excitations [9].

# 4  Computational details

Ground-state properties of the $h$-BN mono- and bilayer and of $MoS_2$ monolayer are calculated within the density functional theory (DFT) using the Quantum-Espresso code [29]. We employ the Perdew-Burke-Ernzerhof (PBE) functional [30] with scalar-relativistic optimized norm-conserving pseudopotentials from the PseudoDojo repository (v0.4) [31] for the $h$-BN and from Fritz-Haber Institute [32] for $MoS_2$. The Kohn-Sham Hamiltonian is diagonalized for a given number of states (see **k**-points and bands $N_b$ in Table 1). These states are used as basis set to represent all operators that enter in Eq. (1). All real-time simulations are carried out using the Yambo code [21]. The EOMs [Eq. (1)] are propagated using the Crank-Nicolson integrator with a time-step of 0.01 fs. In order to take dephasing into account, sampling was taken between 50-200 fs in all simulations. The static dielectric function $\epsilon_{\mathbf{G},\mathbf{G}'}$ that enters in the calculation of screened-exchange self-energy, $\Sigma_{SEX}$ in Eq. (4), is calculated within the random-phase approximation (see Ref. [21] for more details). All calculations are performed in a supercell, so for each system, the susceptibility extracted from the time-dependent polarization is rescaled to the effective thickness, $d_{eff}$, resulting in $\chi_{rescaled}(\omega) = L_z/d_{eff} \cdot \chi(\omega)$ where $L_z$ labels the $z$ dimension of the supercell. To get the SFG and DFG spectra, we carry out simulations for all frequency couples, $\omega_i, \omega_j$, in the desired energy ranges. We use the YamboPy code [33] to extract the relevant susceptibilities from the resulting polarizations [Eq. (6)], as detailed in Sec. 5. All parameters that enter in the different parts of the simulations are reported in Table 1.

# 5 Results

We apply the approach outlined in Secs. 2-3 to the SFG and DFG in $h$-BN and MoS$_2$ mono-layers (Secs. 5.1-5.2). $h$-BN monolayer is a wide band gap insulator with strong excitonic features and provides a clear example of the need for an accurate inclusion of excitonic effects. MoS$_2$ monolayer is one of the most widely studied 2D material, including its SFG/DFG [4, 7]. Since the $h$-BN and MoS$_2$ monolayers belong to the $D_{3h}$ point group [34], they have only one non-vanishing second-order susceptibility tensor element with $\chi_{222}^{(2)} = -\chi_{211}^{(2)} = -\chi_{112}^{(2)} = -\chi_{121}^{(2)} \equiv \chi^{(2)}$.[3] For SFG and DFG, we show results in a heatmap, in which each point has been obtained by a separate real-time simulation. As a guide to reading such heatmaps (see e.g. Fig. 4), we can refer to Eqs. (7)-(8). The susceptibilities corresponding to SFG/DFG have poles both at one-photon ($\omega_{1,2} = \Omega_\lambda$) and at two-photon ($\omega_1 \pm \omega_2 = \Omega_\lambda$) resonances with single-particle transitions (or with excitons) in the system. The one-photon resonances correspond to vertical ($\omega_1$) and horizontal ($\omega_2$) lines in the SFG and DFG heatmaps. In the SFG heatmap, the two-photon resonances correspond to negative slope lines running from $\omega_2 = \Omega_\lambda$ to $\omega_1 = \Omega_\lambda$, while in DFG they correspond to positive slope lines starting at $\omega_2 = \Omega_\lambda$ and $\omega_1 = \Omega_\lambda$. Further, the diagonal $\omega_1 = \omega_2$ corresponds to the SHG in the SFG and the optical rectification in the DFG heatmap. In Sec. 5.3, we consider THz-induced second-harmonic generation in 2L-$h$-BN. This system has inversion symmetry and thus has zero SHG at zero-field.

## 5.1 SFG and DFG in $h$-BN monolayer

In Fig. 4 we report the SFG and DFG spectra for $h$-BN both at the IPA and TD-aGW approximation level. For SFG, the line $\omega_1 = \omega_2$ corresponds to the SHG, already calculated in Ref. [14]. At the IPA level, the SFG spectrum (Fig. 4(a)) of $h$-BN is dominated by two-photon resonances with single-particle transitions between 4-6 eV (negative slope bands). In particular, the lowest-energy band corresponds to the transition from the valence band minimum to the conduction band maximum. The one-photon transitions (vertical and horizontal bands) are much weaker though a significant enhancement is observed for part of the spectra both resonant with one- and two-photon absorption. In particular, this portion of the SFG spectrum is twice as intense as the SHG (the $\omega_1 = \omega_2$ diagonal). The DFG spectrum (Fig. 4(b)) is dominated by the resonant optical rectification between 4-6 eV ($\omega_1 = \omega_2$). Further, one-photon resonances (vertical and horizontal bands) are visible, again enhanced when two-photon resonances are also present. These results can be straightforwardly related to the IPA absorption spectrum (see e.g. Ref. [14], which presents a broad peak between 4-6 eV). This broad peak can be found in the calculated absorption spectrum in Fig. 5(a).

As expected, the addition of the electron-hole interaction drastically changes the SFG/DFG spectra.[4] Sharper and much stronger (note the different color scale) features appear in the TD-aGW spectra, corresponding to the $E_1, E_2$ excitonic peaks at around 6.1 eV and 7 eV [36]. These two excitonic peaks can also be identified in the calculated absorption spectrum at TD-aGW level in Fig. 5(a). Two-photon resonances with the two excitons are clearly visible in the SFG spectrum (negative slope lines in Fig. 4(c)) and DFG spectrum (positive slope lines starting at the exciton energy in Fig. 4(d)). One-photon resonances are also visible (as vertical/horizontal lines). The responses are significantly enhanced in the SFG/DFG when one laser is resonant with $E_1$ ($E_2$) and the second laser with $E_2 - E_1$. These spectral features correspond to the first and third terms in Eq. (8) and provide a measure of the strength of exciton-exciton transitions.

---

[3]For MoS$_2$ we have chosen the orientation of the $x$ axis aligned along an armchair direction following the conventions of Ref. [35].

[4]Note that the shift of the onset is due to the addition of the scissor operator in Table 1 partially compensated by the exciton binding energy.

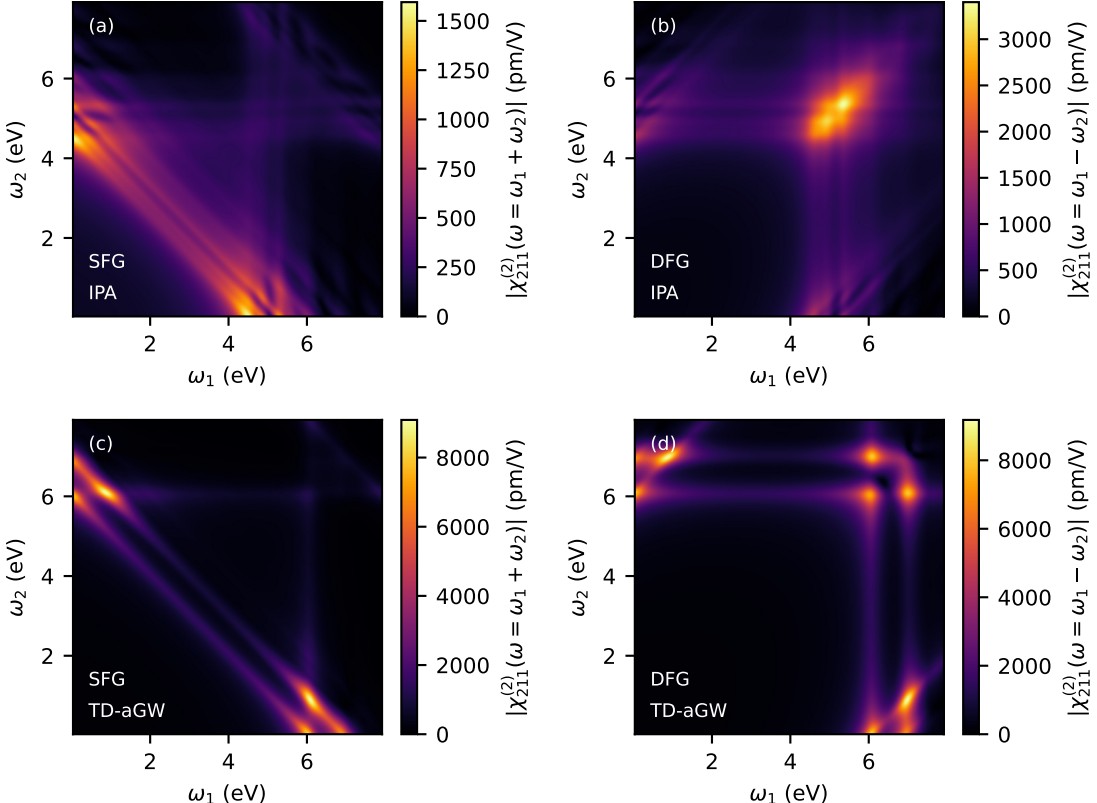

Figure 4: SFG/DFG spectra for $h$-BN in panels (a), (b) at the independent particle level and in panels (c), (d) at the TD-aGW level. The heatmaps have been generated using a frequency grid of $\omega_1 \times \omega_2 = 96 \times 96$ points. For each frequency pair a real-time simulation was run and the output signal processed.

In the DFG spectrum, strong features are visible as well on the diagonal, corresponding to the exciton-resonant optical rectification, as well as when one laser is resonant with one exciton and the second with the other (corresponding to the second and fourth term in Eq. (8)).

The results presented here on the $h$-BN monolayer are a proof of concept of our methodology, showing that resonances with strongly bound excitons are important in both SFG/DFG spectra. Due to the large band gap of $h$-BN, the region of interest for SFG/DFG is difficult to sample and to our knowledge there are no experimental measurements.

## 5.2 SFG and DFG in MoS₂ monolayer

In Fig. 6, we report the SFG and DFG spectra for the MoS$_2$ monolayer at the IPA and TD-aGW level. When compared with $h$-BN, the differences between the IPA and TD-aGW spectra are less striking, as already observed for the absorption [37] and the SHG [14]. Similarly to what was observed for the SHG [14, 38, 39], a significant enhancement is seen at resonances with the C exciton ($\approx 3$ eV), while the weaker A and B excitons ($\approx 2.2$ eV, as spin-orbit coupling is not included the peaks are degenerate) show minimal excitonic enhancement. These excitonic peaks are also visible in the linear absorption spectrum as shown in Fig. 5(b). The SFG spectrum shows a strong two-photon resonance with the C exciton (negative slope line) while the DFG spectrum shows a strong one-photon resonance (vertical/horizontal line) and a strong exciton resonant optical rectification. Though less evident than for $h$-BN, there is an enhancement in the intensity in correspondence of exciton-exciton transitions. In the SFG (Fig. 6(c)),

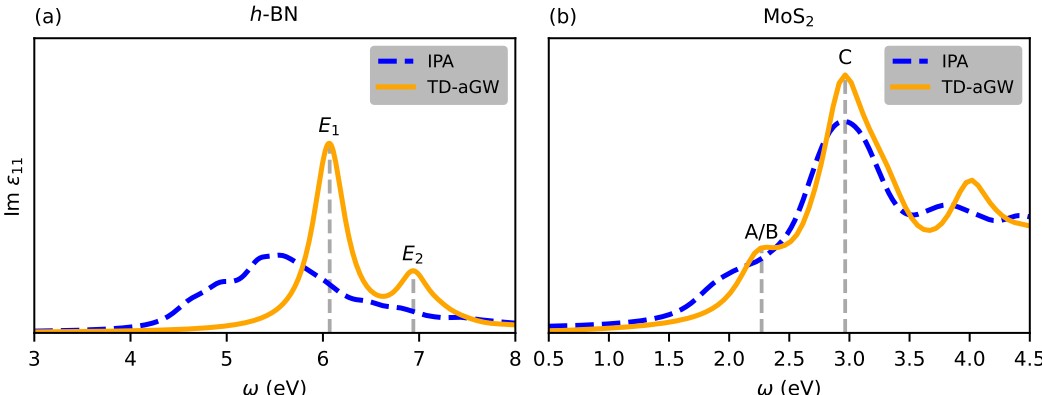

Figure 5: Calculated imaginary dielectric function of (a) *h*-BN and (b) at the IPA (blue dashed line) and TD-aGW (orange solid line) level. The excitonic peaks $E_1$ and $E_2$ of *h*-BN are located at around 6.1 eV and 7 eV, respectively (panel (a)). The degenerate excitonic peak A/B as well as C can be seen at around 2.2 eV and 3 eV, respectively (panel (b)).

the signal is enhanced when one laser is resonant with the A/B exciton (about 2.2 eV) while the frequency of the second matches the energy difference between the C and A/B excitons. In the DFG (Fig. 6(d)), the signal is enhanced when one laser is resonant with the C exciton (about 3 eV) while the frequency of the second corresponds to the energy difference between the C and A/B excitons.

SFG and DFG were measured in mono- [5, 7, 40, 41] and few-layers [4] MoS$_2$. In the presence of metal substrates [40], excitonic resonances were shown to be strongly attenuated, and their position shifted due to gap renormalization. These effects are beyond the methodology presented in this manuscript. Other measurements are performed in a pump-probe configuration with a delay between the pump and probe [40], which in our case requires the inclusion of dephasing effects that are beyond the scope of the present work. Finally, for insulating substrates and synchronized pump and probe, our simulation results are in agreement with existing measurements. In particular, SFG has been observed [41] at 2.9 eV when laser fields at 1.2 eV and 1.9 eV were injected. The DFG was observed [7] by fixing one laser at 3.06 eV and varying the second between 0.79 and 0.95 eV, showing an enhancement of the signal between 2.1-2.2 eV, which the authors attributed to excitonic effects in this region. Our results support this interpretation.

## 5.3 FI-SHG in *h*-BN bilayer

In Fig. 7 we report the third-order susceptibility corresponding to the FI-SHG in the *h*-BN bilayer. For each frequency $\omega$, we run a real-time simulation in the presence of the THz pump ($\nu = 10$ THz). The response functions reported in the figure correspond to two possible experimental configurations, a pump and probe in the $y$ direction ($\chi^{(3)}_{2222}$), and a pump in the $x$ and probe in $y$ direction ($\chi^{(3)}_{1221}$). In both configurations, susceptibility shows a strong resonance at half of the gap, around 2-3 eV, similar to the standard SHG in *h*-BN monolayer [14]. We found that the intensity of the response when the pump and probe are parallel is higher than that in the perpendicular configuration. The order of magnitude of the $\chi^{(3)}$ is comparable with that of bulk ferroelectric oxides which are known for their excellent nonlinear properties [42]. This result implies that two-dimensional crystals can be used as a detector for THz radiation [43].

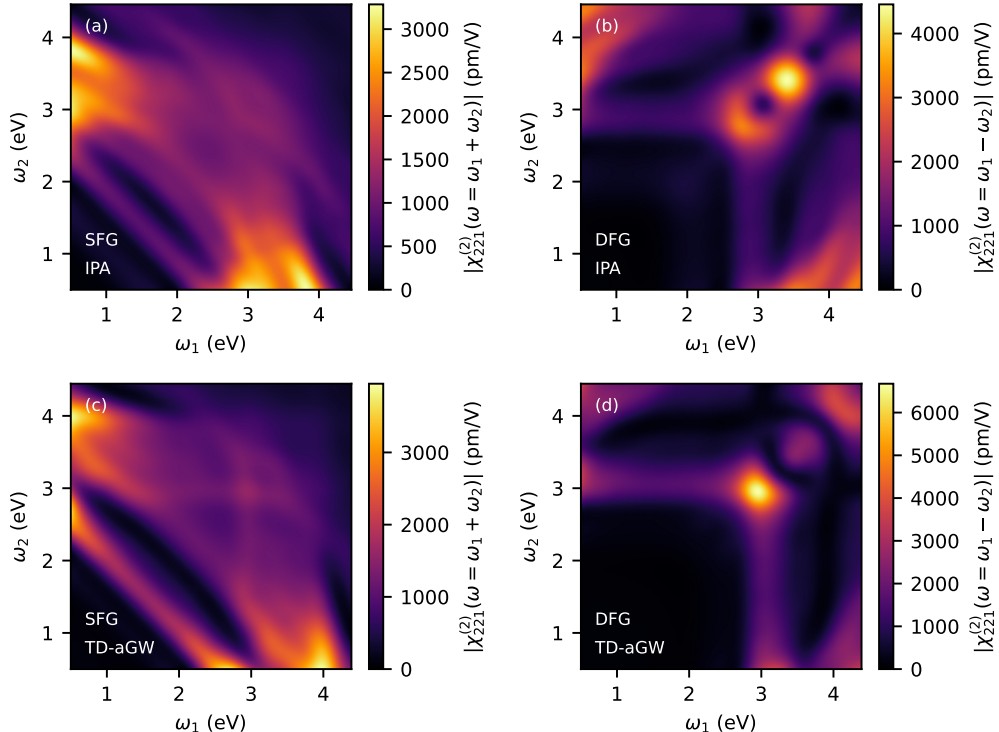

Figure 6: SFG/DFG spectra for MoS$_2$ in panels (a) and (b) at the independent particle level and in panel (c), (d) at the TD-aGW level. The heatmaps have been generated using a frequency grid of $\omega_1 \times \omega_2 = 96 \times 96$ points for the IPA and $\omega_1 \times \omega_2 = 72 \times 72$ for the TD-aGW. For each frequency pair a real-time simulation was run and the output signal processed.

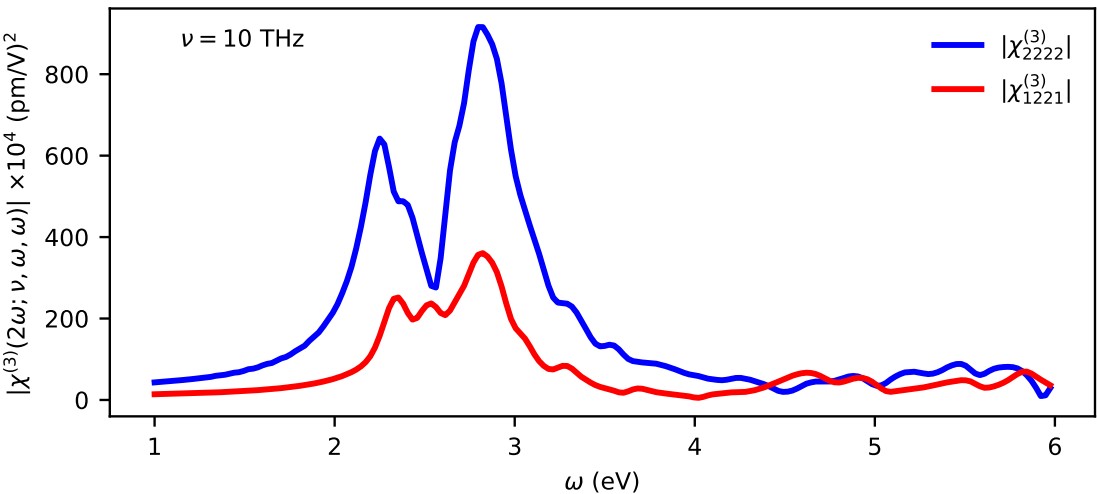

Figure 7: Calculated THz field-induced second-harmonic generation of the $h$-BN bilayer with $\nu$=10 THz at the IPA level. Each curve consists of 192 frequency steps between $\omega = 1-6$ eV.

# 6  Conclusions

In this work, we present a computational framework to study sum/difference frequency generation by means of real-time simulations in the presence of multiple laser fields.[5] With multiple fields, the challenge is the signal processing required to extract the nonlinear susceptibilities. In particular, using a discrete Fourier transform approach may require very long and thus computationally costly simulations. We found that approaches based either on the singular value decomposition and the least squares optimization give accurate results with short sampling time and allow to reduce significantly the simulation time. These approaches enable the calculation of second-order response functions, such as SFG/DFG, and higher nonlinear response functions, as FI-SHG, including excitonic effects within many-body theory. For the studied systems, $h$-BN and $MoS_2$ monolayer, we showed that including excitonic effects in SFG/DFG spectra is critical. In both materials, we predict strong features corresponding to exciton transitions, as recently experimentally observed for another layered material [9]. Further, the results on FI-SHG for the $h$-BN bilayer demonstrate that the approach can be used to predict and interpret nonlinear terahertz spectroscopy of solids [45]. Finally, the presented approach can be coupled to atomic vibrations using finite displacement methods [46,47] opening the way to simulate other spectroscopic techniques such as coherent anti-Stokes Raman spectroscopy (CARS). The latter is a powerful nonlinear optical technique for probing vibrational modes in molecular and solid-state structures. CARS involves two laser beams exciting a vibrational state, and a third beam generating a coherent anti-Stokes signal, allowing for high resolution imaging [48]. CARS can be seen as a combination of SFG and DFG processes, and therefore the method shown in this study could be employed to study the nonresonant CARS response. Since the pure nonresonant CARS is not directly available experimentally [49], the *ab initio* real-time simulation is a promising feature to support CARS measurements.

## Acknowledgments

Calculations for this research were conducted on the Lichtenberg high-performance computer of the TU Darmstadt and at the Höchstleistungsrechenzentrum Stuttgart (HLRS). The authors furthermore acknowledge the computational resources provided by the HPC Core Facility and the HRZ of the Justus-Liebig-Universität Gießen. C.A. acknowledges B. Demoulin and A. Saul for the management of the computer cluster *Rosa*.

**Funding information**  The authors gratefully acknowledge financial support from the Deutsche Forschungsgemeinschaft through the FOR5044 research group (ID: 426703838; http://www.For5044.de). C.A. acknowledges the French "Agence Nationale de la Recherche (ANR)" for financial support (Grant Agreement No. ANR-22-CE30-0027). C.A. and M.G. acknowledge funding from European Research Council MSCA-ITN TIMES under grant agreement 101118915 and from AMUtech. M.G. acknowledges funding from the UKRI Horizon Europe Guarantee funding scheme (EP/Y032659/1).

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
