# Peer review of "A real-time approach to frequency-mixing spectroscopies: application to sum and difference frequency generation in two-dimensional crystals"

_SciPost Physics, doi:SciPost Phys. 19, 129 (2025)_

## Round 1 · Referee Report · Anonymous (Referee 2) · 2025-8-14

Report

The authors present a computational approach for the calculation of two second-order nolinear processes: sum frequency generation (SFG) and difference frequency generation (DFG). The approach is also shown to be suitable for the calculation of field-induced second-harmonic generation (FI-SHG). The authors tackle this challenging problem via real-time simulations and build upon their experience gained in a series of previous articles on the simulation of non-linear and time-dependent phenomena, where they have proven the stability of their time-dependent approach (using the Crank-Nicolson approach) both on the independent-particle level and including electron-hole effects.

In the present manuscript, the authors adress the question how to efficiently extract the relevant components of the second-order susceptibility tensor from the time-dependent polarization that is obtained from the real-time propagation scheme. The standard option here would be discrete Fourier transform which would, however, require long time propagation in order to reliably capture frequency sums and differences. The authors explore two alternative ways: (i) calculating the Moore-Penrose inverse of the matrix that determines the coefficients (when the polarization is developed into harmonics) using singular value deconposition and (ii) performing a least-square fit of the coefficients, minimizing the deviation of the calculated and approximated polarization on a number of time-steps within the simulated time interval. It turns out that this latter method (least-square fit - LSF) works particularly well and allows to reduce the overall time interval for the simulation considerably.

The computational approach is tested on two prototype 2D materials: hBN and MoS2. Unfortunately, there is no comparison with experimental data due to the absence of experimental data for hBN and the limited availability of data for MoS2 (partially measured on metallic substrate and thus not directly comparable to the simulations of the manuscript). Nevertheless, the simulations constitute a valuable proof of concept, in particular through the exploration and validation of the three computational schemes for the extraction of the frequency information.

The manuscript is very clearly written. The results constitute a valuable contribution to the field of computational non-linear optical spectroscopy. Clearly, this manuscript is suitable for publication in SciPost Physics.

Requested changes

(1) If the authors would like to improve the discussion of their results on hBN and MoS2, they might consider to add results on the band-structures and on the (linear) optical spectra. Even though these are pretty well known in the literature, they could be used to visualize the discussion of the computational results. This might improve the readibility and clarity of the second-order results in this manuscript.

(2) Apart from this, I have only a few typos that should be corrected: - page 3, second line from bottom: "can be accounted for" (insert "be") - page 5, first line: alpha should have two subscripts - page 5, six lines below Eq. (6): "there may also resonance" (a "be" seems to be missing in the sentence) - page 5, bottom: DFT is the standard acronym for "density functional theory". Using it for "discrete Fourier transform" as well, might cause confusion. - page 6, caption of Fig. 2: "logarithmically sampled" - page 7, 5 lines below Eq. (11): "As the frequencies of the external fields are muliples of ..." (no komma here) - page 7, 6 lines below Eq. (11): no "then" at the beginning of the line - page 10, bottom paragraph: The phrase "In Fig. 4, we report ..." is doubled. - page 12, line 4 of section 5.2: "Similar to what was observed ..." (insert "was") line 5: "... while the weake A and B excitons ..." (delete "the")

Recommendation

Publish (easily meets expectations and criteria for this Journal; among top 50%)

  • validity: high
  • significance: high
  • originality: high
  • clarity: high
  • formatting: excellent
  • grammar: excellent

Author:  Claudio Attaccalite  on 2025-09-09  [id 5797]

(in reply to Report 1 on 2025-08-14)
Category:
reply to objection

see attachment

Attachment:

Answer_referee_2.pdf

---

## Round 1 · Referee Report · Anonymous (Referee 3) · 2025-8-20

Report

This work uses real-time simulations of two-dimensional crystals in order to study sum and difference frequency generation as well as second-harmonic generation.

The main contribution of this work is a signal processing strategy that is improved with respect to the previous work Ref. [11] by the same team. Consequently, the core part of the manuscript is its section 3. Given that this is incremental technical progress, I would find the work better placed in SciPost Physics Core.

That being said, there is also the application to monolayer $h$-BN and MoS$_2$ and second-harmonic generation in bilayer $h$-BN. In this context, I unfortunately have an issue with the underlying Eq. (2). According to the Hohenberg-Kohn theorem, ground states for a many-body system can be described by a density functional. While I am aware that it is common practice to apply this also to band structures, this is already starting to stretch things. Once one gets to time-dependent density-functional theory (TDDFT) -which is basically the situation investigated here- things get more complicated. Thus, as someone who is not an expert in the physics of excitons, I cannot help wondering if the additional terms in Eq. (2) really go into the right direction from the independent-particle picture. If the authors could add a few further remarks to justify this approximation, this could make the difference to render the manuscript publishable in SciPost Physics.

There are a number of relatively minor details (some of which were also noted by Referee 2) that I list among "Requested changes".

Requested changes

1- Provide some justification of Eq. (2).

2- The acronym "DFG" carries two meanings in the present manuscript. I thus recommend that the authors check if they cannot find better acronyms than "SFG" and "DFG".

3- Likewise, "DFT" is usually understood to mean "density-functional theory" and this method is actually used in the present manuscript. Again, I recommend looking for a different abbreviation of "discrete Fourier transform" in order to avoid confusion.

4- Fig. 2 is referenced and discussed only after Fig. 3. I thus recommend moving it to a later place (where it belongs).

5- The end of the first paragraph of section 3.2.1 refers to Fig. 3. However, I see no logarithmic sampling in Fig. 3. Do the authors maybe mean Fig. 2?

6- There is a number of minor linguistic and typographic issues. I uploaded an annotated manuscript for the authors to help them with proofreading it.

Attachment

Recommendation

Accept in alternative Journal (see Report)

  • validity: high
  • significance: good
  • originality: good
  • clarity: high
  • formatting: excellent
  • grammar: good

Author:  Claudio Attaccalite  on 2025-09-09  [id 5798]

(in reply to Report 2 on 2025-08-20)

see attachment

Attachment:

Answer_to_referee3.pdf

---

## Round 1 · Referee Report · Anonymous (Referee 4) · 2025-8-25

Strengths

1- technically very sound, theory is state-of-the-art 2- studied nonlinear processes interesting and relatively uncharted

Weaknesses

1- Maybe a bit too technical for a general audience

Report

I believe the manuscript is an important contribution in the field of nonlinear response of materials. The studied effects are relatively unknown, and the level of the developed theory is clearly cutting edge, as it includes many-body excitonic effects that play an important role in 2D materials like the ones studied in the manuscript. Therefore, I recommend the publication in SciPost Physics.

Requested changes

1- For clarity, it would be nice if the authors could choose a different letter from “P”, or at least its calligraphy, to denote polarization and momentum, otherwise the derivations are hard to follow at times. This is specially true in sections like 3.1 (page 6); right after Eq. 9, the authors refer to “complex Fourier components \vec{p}_n”, I assume that they want to refer to the quantity defined in Eq. 9 “\vec{p}^(n)”, but the symbol they employ actually corresponds to single-particle momentum introduced above Eq. 6.

2- The description on how the FI-SHG is computed at small finite frequency could be improved. According to Eq. 8, it appears that the response at +-2w is proportional to the third-order susceptibility at frequency set(\nu,\omega\omega). Then, I do not entirely understand the following sentence “summing the χ(3) extracted by P (2ω +) and P (2ω −), one obtains the corresponding FI-SHG for low-frequency time-dependent pump fields.” Do the authors mean that when \nu is positive (negative) it corresponds to 2w+ (2w-)? I believe this should be clarified.

A further minor point: the authors say “Among higher-order responses that can be extracted from Eq. (4), we look at the FI-SHG”, but Eq. 4 explicitly discards higher-than-quadratic contributions. Could they clarify?

3- Why do the authors employ the approximate equality symbol in Eq. 5? To my understanding, this is the exact expression for the second-order susceptibility in perturbation theory.

4- The authors could provide more details on the symmetries of the two studied systems, e.g. their point group, and possibly provide an appropriate reference containing more details on the systems to help the interested reader.

5- The authors rescale the response by a effective thickness d_eff. Is this procedure standard for quadratic responses? Do they expect any non-trivial dependence on this parameter?

Recommendation

Publish (meets expectations and criteria for this Journal)

  • validity: top
  • significance: high
  • originality: high
  • clarity: good
  • formatting: good
  • grammar: good

Author:  Claudio Attaccalite  on 2025-09-09  [id 5799]

(in reply to Report 3 on 2025-08-25)

see attachment

Attachment:

Answer_to_referee4.pdf

---

## Round 1 · Author Response

Dear Editor

Thank you for sending us the referee's report, which we found very positive.
As suggested, we have responded directly to the referee's comments/observations,
and have resubmitted a revised version of our manuscript, which we now consider more suitable for publication in SciPost.

Below is the list of changes made to the manuscript, also highlighted in red in the text.

Kind regards.
Claudio Attaccalite

---

## Round 1 · List of Changes

Here the list of changes in the new version of our manuscript:

1) We updated Ref.31 and Ref. 34 that are now published

2) We added a note to explain that TD-aGW was called TD-HSEX/TD-SEX in previous papers

3) Fixed all abbreviations in references

4) fixed typos

5) We condensed the discussion about problematically long periods at the beginning of Sec. 3.2

6) We changed the introduction of the paragraph about two similar frequencies in Sec. 3.2.1

7) We changed Fig. 3. In the full DFT approach, we used 435 fs sampling time after 40 fs dephasing. For a fair comparison with the other approaches where a 50 fs dephasing is applied, we start also with a 50 fs dephasing in the full DFT approach. Then a simulation time of 385 fs is considered converged. Furthermore, the difference for SVD and LSF in logarithmic scale with 5 fs less sampling time is add to show convergence.

8) We explained why SVD and LSF provide same results and that they are equally efficient according to our tests.

---

## Round 2 · Referee Report · Anonymous (Referee 4) · 2025-9-16

Report

The authors have correctly addressed all my questions. I therefore recommend acceptance for publication in SciPost Physics in its present form.

Recommendation

Publish (meets expectations and criteria for this Journal)

---

## Round 2 · Referee Report · Anonymous (Referee 3) · 2025-9-21

Report

The authors have improved their manuscript and provided a convincing answer to the concerns that I raised previously. I am thus glad to now recommend publication in SciPost Physics.

Requested changes

There are still a number of minor typographic, spelling, or grammatical issues that should be corrected during the production stage.

A recurrent issue is "$h$-BN bilayer". I believe that there are two grammatically correct variants: either "the $h$-BN bilayer" or "bilayer $h$-BN", but these have slightly different meanings. The authors should decide which one corresponds to their intentions and fix this accordingly. The same is true for "$h$-BN and MoS$_2$ monolayer" on page 11.

There are a few further minor details that are noted in the attached PDF.

Attachment

Recommendation

Publish (easily meets expectations and criteria for this Journal; among top 50%)

---

## Round 2 · Referee Report · Anonymous (Referee 2) · 2025-9-30

Report

The authors have responded to my comments. I suggest to go ahead with the publication of the manuscript.

Recommendation

Publish (easily meets expectations and criteria for this Journal; among top 50%)

---

## Editorial Decision

published